# How Robust are the Estimated Effects of Nonpharmaceutical Interventions against COVID-19?

**Mrinank Sharma,**[1,2]* **Sören Mindermann,**[3]* **Jan M. Brauner,**[3]*
**Gavin Leech,**[4] **Anna B. Stephenson,**[5] **Tomáš Gavenčiak,**
**Jan Kulveit,**[6] **Yee Whye Teh,**[1] **Leonid Chindelevitch,**[7] **Yarin Gal**[3]

[1] Department of Statistics, University of Oxford, UK.
[2] Department of Engineering Science, University of Oxford, UK.
[3] OATML Group, Department of Computer Science, University of Oxford, UK.
[4] Department of Computer Science, University of Bristol, UK.
[5] School of Engineering and Applied Sciences, Harvard University, USA.
[6] Future of Humanity Institute, University of Oxford, UK.
[7] MRC Centre for Global Infectious Disease Analysis;
and the Abdul Latif Jameel Institute for Disease and Emergency Analytics (J-IDEA),
School of Public Health, Imperial College London.

## Abstract

To what extent are effectiveness estimates of nonpharmaceutical interventions (NPIs) against COVID-19 influenced by the assumptions our models make? To answer this question, we investigate 2 state-of-the-art NPI effectiveness models and propose 6 variants that make different structural assumptions. In particular, we investigate how well NPI effectiveness estimates generalise to unseen countries, and their sensitivity to unobserved factors. Models that account for noise in disease transmission compare favourably. We further evaluate how robust estimates are to different choices of epidemiological parameters and data. Focusing on models that assume transmission noise, we find that previously published results are remarkably robust across these variables. Finally, we mathematically ground the interpretation of NPI effectiveness estimates when certain common assumptions do not hold.

## 1 Introduction

Nonpharmaceutical interventions (NPIs), such as business closures, gathering bans, and stay-at-home orders, are a central part of the fight against COVID-19. Yet it is largely unknown how effective different NPIs are at reducing transmission [2, 7]. With a global rise of COVID-19 cases and an unknown number of waves to come, better understanding is urgently needed to guide policy. Indeed, knowing the effectiveness of different NPIs would enable countries to efficiently suppress the disease without imposing unnecessary burden on the population.

Data-driven NPI modelling is one of the best approaches for inferring NPI effect sizes. These models assume that the implementation of an NPI affects the course of a country's epidemic in a particular way. Then, using publicly available incidence and fatality data, as well as a list of NPIs with their implementation dates, the NPI model can be inverted, yielding NPI effectiveness estimates.

---

Correspondence to <mrinank@robots.ox.ac.uk>, <soren.mindermann@cs.ox.ac.uk>,
<jan.brauner@cs.ox.ac.uk>.

However, it is impossible to construct a model without making assumptions. Given the importance and the policy-relevance of NPI effectiveness estimates, we must ask *to what extent are our NPI effectiveness estimates influenced by the assumptions our models make?* If our estimates fluctuate widely under different plausible assumptions, our results cannot be used to inform policy.

Additionally, it is challenging to collect data about NPI implementation dates in several countries, so all analyses are limited to a subset of countries. Also, epidemiological parameters describing COVID are required by NPI effectiveness models, but they are only known with uncertainty. In order for effectiveness estimates to be used by policymakers, we must also assess their robustness to these factors.

To address these challenges, we empirically investigate the influence of common assumptions made by NPI effectiveness models. We build on previous state of the art NPI effectiveness models [2, 7] and construct 6 variants that make different structural assumptions. Without access to ground-truth NPI effectiveness estimates, we evaluate models by assessing how well their estimates generalise to unseen countries, and how much their estimates are influenced by unobserved factors. We find that assuming *transmission noise* yields more robust estimates that also generalise better.

Furthermore, we systematically validate all of our models, assessing how sensitive NPI effectiveness estimates are to variations in the input data and assumed epidemiological parameters. We find that systematic trends in effectiveness estimates obtained from our models when varying model structure, data, and epidemiological parameters. In particular, *closing schools and universities in conjunction* was consistently highly effective; the effect size of *stay-at-home orders* is modest; the additional benefit of closing most nonessential businesses was smaller than targeted closures of high exposure businesses; and the effectiveness of gathering bans increased as the maximum gathering size decreased. Our model implementations and sensitivity analyses can be found at https://github.com/epidemics/COVIDNPIs/tree/neurips.

Finally, we mathematically ground the interpretation of NPI effectiveness estimates when common assumptions do not hold. In particular, we conclude that our estimates should be interpreted as *average, marginal* effectiveness estimates, where the average is taken over the situations in which each NPI was active. As such, we urge caution in interpreting results from data driven NPI effectiveness models. For instance, mask-wearing mandates for (some) public spaces were only activated in our data when several other NPIs were also activated. Therefore, we can only reason about the effectiveness of mask-wearing mandates in the presence of many other NPIs.

We hope that these results will advance understanding and best practices of COVID-19 NPI effectiveness models, ultimately helping countries efficiently suppress virus transmission.

**Disclaimer.** Note that this paper uses the same data as our previous work [2] (`medRvix Version` 4) and references previously reported results in several places. While our latest results use an updated dataset and model, the results in this paper have not been updated. The difference does not affect the claims made here. The exact models and data used to produce these results can be found on Github.

## 2 Common assumptions in NPI modelling

To investigate the influence of specific assumptions, we must first understand why these assumptions are made. To infer NPI effect sizes, data-driven NPI effectiveness models must somehow link the course of a country's epidemic to NPI implementation dates. Fig. 1 broadly outlines the approach that our models take. In short, these models assume that implementing an effective NPI immediately reduces transmission of COVID-19. This transmission is measured using the *reproduction number*, $R$. $R$ is the expected number of people directly infected by one infected person. Therefore, given a list of NPIs, their effectiveness estimates, and an estimate of the transmission that occurs when no NPIs are active (the *basic reproduction number*, $R_0$), we can compute $R$ on a specific day.

However, $R$ is insufficient to calculate the number of infections on a particular day; we also need to know the time delay between a person becoming infected and then subsequently infecting $R$ others. This is the *Generation Interval* (GI), but published estimates vary and often depend on the specific country studied. Furthermore, we also need to know the time delay between infection and case/death reporting to link the number of infections to the number of reported cases and deaths (our observations). These time delays are also only known with uncertainty.

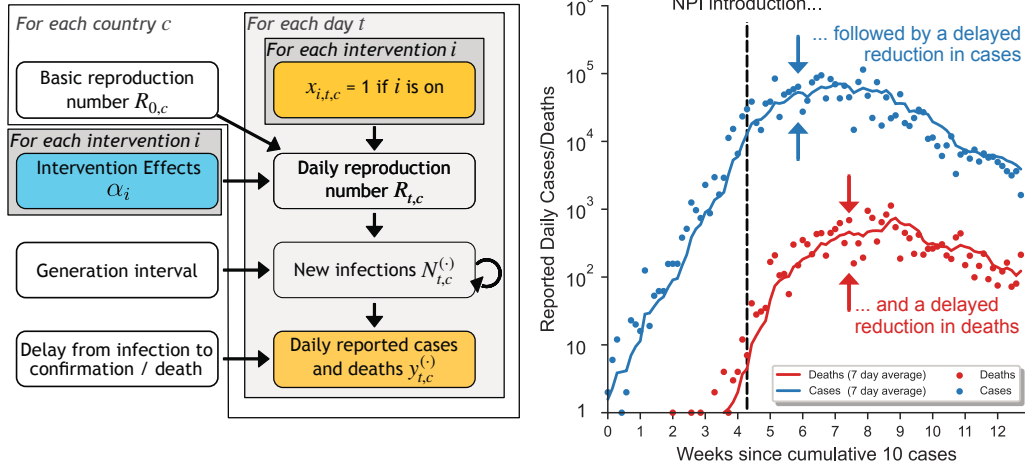

**Figure 1:** Approach Overview. Data-driven NPI effectiveness models assume that interventions affect the course of a country's epidemic. Note: the right subplot shows simulated data with only one NPI. In reality, most countries implemented several NPIs, in different orders.

Without making assumptions, our models would be unable to infer NPI effectiveness estimates. The key question we seek to answer is not whether these assumptions hold in reality (since they do not), but rather the extent to which our results are the product of a particular assumption.

We proceed by collecting and reviewing the assumptions of our previous work [2]. We then propose alternative plausible assumptions that would lead to different models. We discuss key assumptions in this section, but include a more detailed discussion in the Supplement (Section A.5).

**Notation**. We index time with $t$ and country with $c$. The reproduction number $R$ is the expected number of infections caused by one infection (if all members of the population were susceptible). The basic reproduction number for country $c$ (i.e., $R$ in the absence of any observed NPIs) is $R_{0,c}$. The time-varying (instantaneous [8]) reproduction number at time $t$ in country $c$ is $R_{t,c}$, which we use as the measure of transmission. $x_{i,t,c}$ are binary NPI activation features with $x_{i,t,c} = 1$ indicating that NPI $i$ is active in country $c$ at time $t$. $y_{t,c}^{(C)}$ and $y_{t,c}^{(D)}$ represent the number of daily reported cases and deaths respectively. The set of NPIs is denoted as $\mathcal{I}$. $N_{t,c}$ represents (constant-scaled) numbers of new daily infections. $\alpha_i \in \mathbb{R}$ parameterises the effectiveness of NPI $i$ and $\alpha_i > 0$ is interpreted as NPI $i$ being effective. Superscript $(C)$ represents terms corresponding to reported cases, while superscript $(D)$ corresponds to reported deaths.

## 2.1 Default Model Outline

To link NPI implementation dates to reported cases and deaths, our models require knowledge of COVID-19. For example, the generation interval describes the time between successive infection events. Further, we also need to know the delay between infection and case/death reporting i.e., the time delay between a person becoming infected, and their case/death being reported in national statistics. Since these delays vary across countries and over time, they are difficult to estimate. This motivates the following assumption.

**Assumption 1.** Epidemiological parameters are constant across countries and time [7, 1, 3, 22, 28, 2].

We also need to link NPI implementation and effectiveness estimates to our measure of COVID-19 transmission, $R_{t,c}$. This is a challenging task. In reality, NPI effectiveness will vary over time as adherence changes, and will also depend on the specific NPI implementation in a particular country. For instance, some countries required residents to complete a form to leave their home during a *stay-at-home order*, whilst others did not. A common approach to address this is to *pool* estimates across countries and time, which motivates the following assumptions.

**Assumption 2** (Constant NPI Effectiveness). (a) The effectiveness of NPI $i$ is independent of country [7, 1, 3, 28, 2]. (b) The effectiveness of NPI $i$ is independent of time [7, 1, 3, 28, 2].

In addition, the effectiveness of different NPIs may depend on the other active interventions. Social distancing measures may reduce the effectiveness of *mandatory mask wearing*. However, our data is

limited. Since we don't observe all combinations of NPIs in each country, it is challenging to model NPI interactions. Instead, it is common to assume multiplicative, independent NPI effects.

**Assumption 3** (Multiplicative NPI Effects). NPIs have multiplicative effects on $R_{t,c}$ [7, 1, 3, 2].

**Assumption 4** (No NPI Interactions). The effectiveness of NPI $i$ is independent of the other NPIs that are active [7, 1, 5, 2].

Finally, many factors will affect COVID-19 transmission, such as behavioural change and unrecorded interventions. However, our models assume that only observed NPIs influence $R_t$.

**Assumption 5** (No Unobserved Factors). $R_{t,c}$ depends only on $R_{0,c}$ and the active NPIs i.e., $\{x_{i,t,c}\}_{i \in \mathcal{I}}$. Therefore, each NPI has its full effect on $R_{t,c}$ immediately [7, 1, 3, 2].

With Assumptions 2 to 5, we can write:

$$R_{t,c} = R_{0,c} \prod_{i \in \mathcal{I}} \exp(-\alpha_i x_{i,t,c}). \tag{1}$$

$R_{t,c}$ is computed as the basic reproduction number in country $c$ multiplied by country-independent factors, each of which correspond to active NPIs. We are now able to link NPI implementations to the time-varying reproduction number, $R_{t,c}$, for each country.

We now wish to use $R_t$ to compute the number of daily infections. We will use the *discrete time growth rate*, $g_{t,c}$ to do so. $g_{t,c}$ describes the *change* in the number of daily infections, and satisfies $N_{t,c} = g_{t,c} N_{t-1,c}$. In other words, the number of infections of day $t$ in country $c$, is equal to the number of infections on the previous day, multiplied by the growth rate. If $g_{t,c} = 1$, then there is no *change* in the number of infections on subsequent days. How can we link $R_{t,c}$ to $g_{t,c}$?

**Assumption 6.** $R_{t,c}$ may be converted to $g_{t,c}$ by assuming constant exponential growth [32]:

$$g_{t,c} = \exp\left(M_{\mathrm{GI}}^{-1}(R_{t,c}^{-1})\right), \tag{2}$$

where $M_{\mathrm{GI}}^{-1}$ is the inverse moment-generating function of the generation interval distribution [2], [7] (in their sensitivity analysis).

Note that to convert $R_{t,c}$ to a daily growth rate, we required parameters of the *generation interval distribution*: the distribution describing the time between one infection and the subsequent generated infections. For example, if the generation interval distribution was a delta distribution at $t = 5$ days, an infected person would infect $R_t$ others after exactly 5 days.

Since we observe both cases and deaths, we model two sets of daily infection counts. $N_{t,c}^{(C)}$ represents the daily number of infections on day $t$ in country $c$ that will lead to reported cases after a time delay. Similarly, $N_{t,c}^{(D)}$ represents the daily number of infections on day $t$ in country $c$ that will lead to reported deaths after a longer time delay. We also introduce noise on the daily growth rate, $g_{t,c}$ to account for unobserved factors influencing transmission, which partially relaxes Assumption 5 *(No Unobserved Factors)*.

**Assumption 7** (Transmission Noise). (a) There is multiplicative noise on the measure of transmission (usually $g_{t,c}$ or $R_{t,c}$) [2] (similarly used in older epidemic models [8, 29]). (b) In expectation, the measure of transmission is the same for cases and deaths [2].

We can now write:

$$N_{t,c}^{(C)} = N_{0,c}^{(C)} \prod_{t'=1}^{t} \left[ g_{t',c} \cdot \exp\left(\varepsilon_{t',c}^{(C)}\right) \right], \quad N_{t,c}^{(D)} = N_{0,c}^{(D)} \prod_{t'=1}^{t} \left[ g_{t',c} \cdot \exp\left(\varepsilon_{t',c}^{(D)}\right) \right], \tag{3}$$

with noise terms $\varepsilon_{t',c}^{(C)}, \varepsilon_{t',c}^{(D)} \sim \mathcal{N}(0, \sigma_g^2)$. Transmission noise partially relaxes Assumption 5 (*No Unobserved Factors*) as this noise can account for unobserved factors that influence $R$. If the timing of an unobserved factor is uncorrelated with the observed NPIs [5], we expect the unobserved factor to be attributed to noise. However, if unobserved NPI $i$ is correlated with observed NPI $j$, the effect of NPI $i$ may be attributed to NPI $j$. As our NPI effectiveness models operate with many unobserved factors, caution is needed in drawing causal conclusions from such observational studies. We discuss this further in the Supplement A.5.

In addition, this noise can model time-varying changes in the rate of case/death reporting. Specifically, transmission noise allows for time-varying changes in the Ascertainment Rate, $\mathrm{AR}_c$, (the proportion of infected cases that are subsequently reported) and the Infection-Fatality Rate, $\mathrm{IFR}_c$, (the proportion of infected cases that subsequently die) since $\varepsilon_{t',c}^{(C)}$ affects $N_{t,c}^{(C)}$ for all $t \geq t'$ [2]. For example,

if the proportion of infections tested in country $c$ increases by $20\%$ on day $t'$, the model can set $\varepsilon_{c,t'}^{(C)} = \log 1.2 = 0.18$, which will increase *all* future case numbers.

Of course, there are also time-invariant differences in case and death reporting, as well as healthcare quality (that would influence the proportion of infections that die). However, these differences in $\text{IFR}_c$ and $\text{AR}_c$ are accounted for by latent variables $N_{0,c}^{(C)}$ and $N_{0,c}^{(D)}$, which represent infection numbers of the first day of analysis. Concretely, if the true number of infections in country $c$ is the same as country $c'$ for all $t$, but $c$ tests a greater proportion of the population, we may infer $N_{0,c}^{(C)} > N_{0,c'}^{(C)}$.

With the above assumptions, we are able to compute the daily number of infections over a time period if an initial outbreak size, $N_{0,c}$, is provided. We now seek to map infection counts to our observations: reported cases and deaths. The daily infections that are eventually reported, $N_{t,c}^{(C)}$, and the daily infections that eventually result in death, $N_{t,c}^{(D)}$, are convolved with the delays between infection and case/death reporting to produce the expected number of new reported cases $\bar{y}_{t,c}^{(C)}$ and deaths $\bar{y}_{t,c}^{(D)}$:

$$\bar{y}_{t,c}^{(C)} = \sum_{\tau=0}^{31} N_{t-\tau,c}^{(C)} \pi_C[\tau], \quad \bar{y}_{t,c}^{(D)} = \sum_{\tau=0}^{47} N_{t-\tau,c}^{(D)} \pi_D[\tau]. \tag{4}$$

$\pi_C[\tau]$ represents the probability of the delay between infection and case reporting being $\tau$ days, while $\pi_D[\tau]$ represents the probability of the delay between infection and death reporting being $\tau$ days. For computational reasons, we right-truncate these delay distributions at a maximum delay of 31 days (cases) and 47 days (deaths).

**Assumption 8.** The output distribution of (observed) reported cases $y_{t,c}^{(C)}$ and deaths $y_{t,c}^{(D)}$ follows a Negative Binomial (NB) distribution [7, 2, 1]:

$$y_{t,c}^{(C)} \sim \text{NB}(\mu = \bar{y}_{t,c}^{(C)}, \Psi^{(C)}), \qquad y_{t,c}^{(D)} \sim \text{NB}(\mu = \bar{y}_{t,c}^{(D)}, \Psi^{(D)}), \tag{5}$$

where $\Psi^{(C)}$ and $\Psi^{(D)}$ are the dispersion parameters (larger $\Psi$ correspond to less noise) for cases and deaths, which are inferred from the data. The negative binomial distribution is suitable as it has support over $\mathbb{N}_0$, and allows for over-dispersion, with independent mean and variance parameters.

## 2.2 Alternative Assumptions

We now propose alternative assumptions to those of the default model. We later use these assumptions to construct alternative models.

**Additive Effects.** Instead of Assumption 3 (*Multiplicative NPI Effects*), we could assume additive NPI effects.

**Assumption 9** (Additive NPI Effects)**.** The introduction of NPI $i$ has an additive effect on $R_{t,c}$ by affecting a non-overlapping, constant proportion of initial transmission, $R_{0,c}$. The introduction of NPI $i$ eliminates all transmission related to $i$.

This may be intuitively understood as follows. Transmission occurring in the absence of NPIs may be due to non-overlapping fractions. For example, $25\%$ of $R_0$ could be associated with educational institutes, $35\%$ with businesses, and $40\%$ unaffected by NPIs. Closing businesses would eliminate the corresponding $35\%$ of transmission. This leads to:

$$R_{t,c} = R_{0,c} \left( \hat{\alpha} + \sum_{i \in \mathcal{I}} \alpha_i \left( 1 - x_{i,t,c} \right) \right), \quad \text{with } \hat{\alpha} + \sum_{i \in \mathcal{I}} \alpha_i = 1, \tag{6}$$

$\alpha_i > 0 \; \forall i$ and $\hat{\alpha} > 0$. $\alpha_i$ is the proportion of transmission eliminated by introducing NPI $i$ while $\hat{\alpha} > 0$ represents the proportion of transmission that remains even when all NPIs are active.

**Different Effects.** It is possible to simultaneously relax Assumptions 4 (*No NPI Interactions*) and 2a (*Constant NPI Effectiveness over Countries*) by allowing NPI effects to vary across countries. For example, if we find a relatively higher effectiveness for NPI $i$ in country $c$, this could be caused by interactions with the other NPIs that were active in country $c$ when $i$ was implemented, or by other country-specific factors such as country $c$'s population demographics.

**Assumption 10** (Different NPI Effects)**.** Country-specific NPI effectiveness parameters, $\{\alpha_{i,c}\}_c$, are drawn *i.i.d.* according to $\mathcal{N}(\alpha_i, \sigma_\alpha^2)$. $\sigma_\alpha$ is a hyperparameter describing the variance in effectiveness across countries.

**Noisy-R**. Transmission noise could instead be applied directly to $R_{t,c}$ rather than to $g_{t,c}$ (Eq. 3):

$$R_{t,c}^{(C)} = \bar{R}_{t,c} \exp \varepsilon_{t,c}^{(C)}, \quad R_{t,c}^{(D)} = \bar{R}_{t,c} \exp \varepsilon_{t,c}^{(D)}, \tag{7}$$

where $\varepsilon_{t,c}^{(C)}, \varepsilon_{t,c}^{(D)} \sim \mathcal{N}(0, \sigma_R^2)$.

**Discrete Renewal Infection Process.** We previously converted $R_{t,c}$ to $g_{t,c}$ by assuming constant exponential growth (Assumption 6). We can alternatively use a discrete renewal process that does not make this assumption [6, 28]. We then write:

$$N_{t,c}^{(C)} = R_{t,c}^{(C)} \sum_{\tau=1}^{28} N_{t-\tau,c}^{(C)} \cdot \pi_{GI}[\tau], \quad N_{t,c}^{(D)} = R_{t,c}^{(D)} \sum_{\tau=1}^{28} N_{t-\tau,c}^{(D)} \cdot \pi_{GI}[\tau], \tag{8}$$

Under this infection model, transmission noise would be applied to $R_{t,c}$ as in Eq. (7). $\pi_{GI}[\tau]$ is the truncated, discretised generation interval distribution.

**No Transmission Noise.** Recall that transmission noise can be used to explain time-varying changes in reporting and treatment, as well as unobserved factors. Alternatively, *output noise* could be used to model these factors.

**Assumption 11** (No Transmission Noise). There is no noise in the measure of transmission ($R_{t,c}$ or $g_{t,c}$) [7, 22, 1].

## 3 Experiments & Methodology

We now use previously outlined assumptions to construct 8 models that make different structural assumptions. By comparing NPI effectiveness estimates under these models, we effectively compare effectiveness estimates *under different assumptions*. This will allow us to assess how the assumptions made influence our NPI effectiveness estimates.

The models that we construct are: *Default*, the model used our previous work [2] (`medRvix Version 4`); *Additive Effects*, where the NPI interaction is additive; *Different Effects*, where NPI effectiveness is allowed to vary per country; *Noisy-R*, where noise is applied to $R_{t,c}$ rather than $g_{t,c}$; *Discrete Renewal (DR)*, where the infection model is a discrete renewal process and noise is applied on $R_{t,c}$; *Deaths-Only DR* — identical to the *DR* model, except only deaths are modelled; *Flaxman et al. [7]*, which is identical to *Deaths-Only DR*, but has no transmission noise; and *Default (No Transmission Noise)*, which is identical to *Default* but has no transmission noise. Fig. 4 (Supplement) outlines the differences between these models.

**Model Evaluation.** While our models are reasonable *a priori*, we must also empirically validate them. In particular, an analysis of holdout predictive performance is required, even though prediction is not our purpose [11, 12]. Holdout predictive performance can be used to rule out models—since we expect that a significant fraction of variation in national cases and deaths *can* be explained by NPIs, there is little reason to trust an NPI model that entirely fails to predict on held-out data. We measure holdout predictive likelihood on a test-set of 6 countries, having tuned hyperparameters by cross-validation.

In addition, we must assess how NPI effectiveness estimates from these models are influenced by unobserved factors. Our data does not capture all NPIs implemented in each region, and transmission is also influenced by other variables, including behaviour change not attributable to our NPIs. Here, we assess sensitivity to unobserved factors by evaluating how inferred NPI effectiveness parameters change when previously observed NPIs become unobserved (NPI leave-outs), as well as when previously unobserved NPIs are observed [30]. Additional NPIs are drawn from the Oxford COVID-19 Government Response Tracker (OxCGRT) NPI dataset [14]. We favour models with stable effectiveness estimates under these conditions, as this indicates the model assigns unobserved effects to noise and not to our NPIs.

We report sensitivity to unobserved factors using the following loss: $\mathcal{L}_c = \frac{1}{|\mathcal{I}|} \sum_{i \in \mathcal{I}} \text{std}[\text{median}(\tilde{\alpha}_i)]$, where the standard deviation is over multiple test conditions of analysis category $c$ e.g., if $c =$ NPI leave-outs, the standard deviation is taken over several model runs where one NPI at a time is made unobserved and all else is equal. We compute the standard deviation in posterior median NPI effectiveness across tests for every NPI, and then average this over our NPIs. $\tilde{\alpha}_i$ is the effectiveness of NPI $i$ in units "percentage reduction in $R$";. Larger $\mathcal{L}_c$ indicates higher sensitivity.

**Model Robustness.** Finally, following our previous work [2], we perform extensive sensitivity experiments across 6 additional categories for these models. To examine *sensitivity to data* we vary:

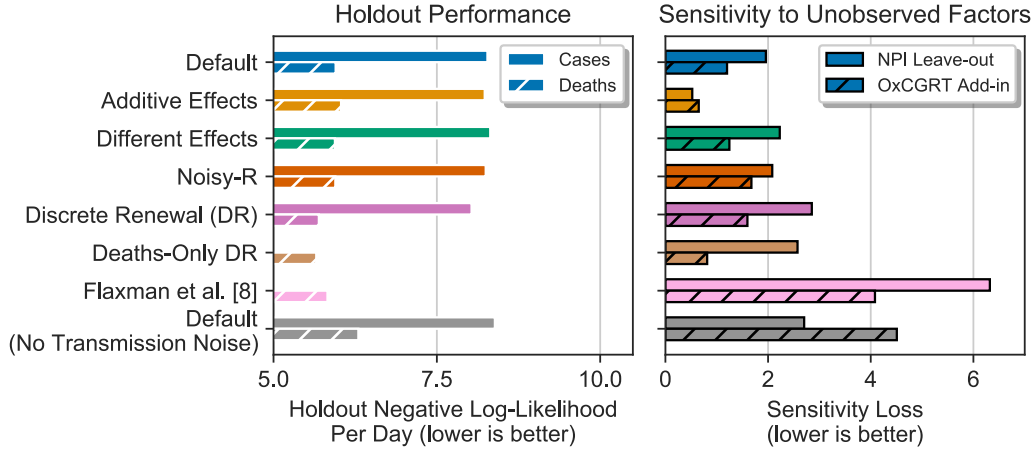

**Figure 2:** Model comparison. *Left*: holdout performance, measured using predictive log-likelihood on a test-set of 6 regions *Right*: sensitivity to unobserved factors.

the included countries, by holding out each country one at a time; the cumulative threshold for case masking; and the threshold for death masking. Recall that all NPI studies are limited to a limited number of countries due to data collection difficulties. Additionally, since epidemiological parameters are only known with uncertainty, we examine *sensitivity to epidemiological parameters* by varying the parameters of the generation interval; the infection to case reporting delay; the infection to death reporting delay distributions; the prior mean over $R_{0,c}$; and the prior over NPI effectiveness. In total, the sensitivity analysis includes over 600 experimental conditions.

**Data & Implementation.** We use our previous [NPI dataset](#) [2], composed of data on the implementation of 9 NPIs in 41 countries between January and end of May 2020 (validated with independent double entry). Data on reported cases and deaths is from the Johns Hopkins CSSE tracker [19]. Please see Supplement A.7.1 for further details. We implement our models in PyMC3 [31], using Hamiltonian Monte Carlo NUTS [15] for inference. We use 4 chains with 1250 samples per chain. For runs with default settings, we ensure that the Gelman-Rubin $\hat{R}$ is less than 1.05 and that there are no divergent transitions. Our sensitivity analyses and model implementations are available [online](#).

**Related Work.** Unfortunately, holdout performance validation is often limited or absent in other previous work. The majority of NPI studies do not report holdout performance [16, 17, 1, 23, 3, 22, 24, 9, 18]. Flaxman et al. [7] hold out the last fourteen days in all countries in parallel. While sensitivity analyses are more common than holdout validation often only a small subset of epidemiological parameters are examined and sensitivity to model structure (structural sensitivity) is not evaluated. Flaxman et al. [7] check sensitivity to the generation interval and to leaving out individual countries, fit the reproduction number ($R$, the expected number of infections directly generated by one infected individual) with a non-parametric model, and compare to an alternative model of $R_0$. Banholzer et al. [1] check the sensitivity of their results to the delay from infection to reporting, the threshold initial case count, influential single data points, the form of the influence function, and to restricting NPI effectiveness to be positive. Jarvis et al. [18] varied the reduction in post-lockdown contact among young people. Many NPI studies do not mention sensitivity or validation at all [3, 28, 4, 20, 26, 24, 9].

## 4 Results & Discussion

Fig. 2 shows holdout predictive performance and sensitivity to unobserved factors for these models. Holdout performance is similar across models, but consistently better for deaths than cases, reflecting that the predictions for cases are for more days and further into the future than for death, as deaths appear later than cases—see also Fig. A.2. However, the sensitivity to unobserved factors varies significantly across models; in particular, the discrete renewal model is more sensitive than the default model. Since the sum of NPI effectiveness estimates is constrained for the *Additive Effects* model, it has the lowest sensitivity. Furthermore, we find that including transmission noise *both* improves holdout predictive performance and increases robustness to unobserved factors, suggesting models with transmission noise are less likely to assign unobserved factors to observed NPIs. Further, transmission noise more closely reflects the underlying stochastic process and has history in epidemic

modelling [8, 29]. Therefore, we proceed by excluding models without transmission noise in subsequent analyses.

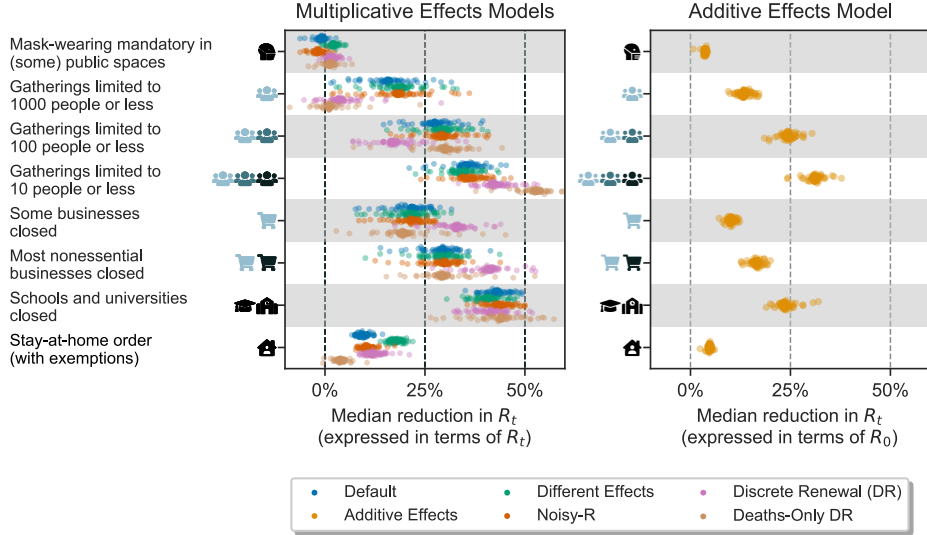

**Figure 3:** Aggregated sensitivity analysis for models with transmission noise. Each dot represents posterior median effectiveness in one experimental condition for one model. Since the *Additive Effects* model expresses NPI effectiveness in a different unit, we show its results in a separate plot.

**Structural sensitivity analysis.** Having excluded models without transmission noise, we now assess the robustness of inferred NPI effectiveness estimates to variations in the data and epidemiological parameters for the remaining 6 models.

Fig. 3 shows the results of our additional sensitivity analyses. The *Additive Effects* Model is plotted separately to reflect that the effectiveness values cannot be directly compared: the multiplicative models represent effectiveness values in terms of multiplicative reductions in $R_t$, while the *Additive Effects* Model represents effectiveness as additive reductions in $R_t$, but as percentages of $R_0$. Consequently, the absolute reduction in $R_t$ when an NPI is implemented is independent of other active NPIs for the *Additive Effects* Model, but not for the multiplicative effects models.

We find systematic trends in median NPI effectiveness estimates, even across variations in model structure, data and epidemiological parameters. *Stay-at-home order* and *mask-wearing mandates* are consistently among the least effective NPIs, suggesting they may have had played a relatively small role in reducing transmission in our window of analysis. *Closing schools and universities in conjunction* tends to be one of of the most effective NPIs (these two NPIs cannot be separated since they are highly collinear–see [2]). Amongst the multiplicative effect models, we find that that marginal benefit of *most nonessential businesses closed*, i.e., the additional reduction in transmission when most nonessential businesses are closed given that some businesses are already closed, is modest. Curiously, the *DR* and *Deaths-Only DR* models both find a relatively lower effectiveness for *gatherings limited to 1000 or less* and a relatively higher effectiveness for *gatherings limited to 10 or less*, but differ substantially in the estimates for *gatherings limited to 100 or less*. We suggest that lower effectiveness of *gatherings limited to 100 or less* for the *DR* model is in part due to the effectiveness of *some businesses closed* being relatively higher. We discuss our results and their potential policy implications in greater depth in our previous work [2].

## 5 Effectiveness Depends on Context

If we do not use Assumption 10, we assume that NPI effectiveness is constant over countries and time, and does not depend on the other active NPIs (Assumptions 4 and 2). In reality, these assumptions do not hold. For instance, *mask wearing mandates* may have a greater effect on $R$ when no social distancing measures are in place. Also, the specifics of NPI implementation differ across countries e.g., some countries required residents to complete a form to leave their home during a stay-at-home order, whilst others did not. Furthermore, NPI adherence (and thus effectiveness) will vary over time.

How should effectiveness estimates be interpreted when these assumptions are violated? To gain insight, we assume that ground truth values of $g_{t,c}$, $R_{t,c}$ and $R_{0,c}$ have been provided. Consider simplified versions of the *Default* Model and the *Noisy-R* Model that directly observe these values.

**Simplified Default Model.** $g_{t,c} = g(R_{t,c}) \exp(\varepsilon_{t,c})$, with $R_{t,c} = R_{0,c} \prod_{i \in \mathcal{I}} \exp(-\alpha_i \, x_{i,t,c})$.
**Simplified Noisy-R Model.** $g_{t,c} = g(R_{t,c})$, with $R_{t,c} = R_{0,c} \exp(\varepsilon_{t,c}) \prod_{i \in \mathcal{I}} \exp(-\alpha_i \, x_{i,t,c})$,

i.e., the *Simplified Default* Model applies noise $\varepsilon_{t,c} \sim \mathcal{N}(0, \sigma^2)$ to $g_{t,c}$ whilst the *Simplified Noisy-R* Model applies noise to $R_{t,c}$. We now derive expressions for the Maximum Likelihood (ML) estimates of $\alpha_i$ given $\{\alpha_j\}_{j \neq i}$, presented in terms of $\exp(-\alpha_i)$ i.e., the factor by which NPI $i$ reduces $R$.

Let $\Phi_i = \{(t,c) | x_{i,t,c} = 1\}$ be the days and countries with NPI $i$ active.
Let $\tilde{R}_{(-i),t,c} = R_{0,c} \prod_{j \in \mathcal{I} \setminus \{i\}} \exp(\alpha_j x_{j,t,c})$ i.e., $\tilde{R}_{(-i),t,c}$ is the predicted $R$ ignoring the effect of $i$.

**Theorem 1.** The ML estimate of $\alpha_i$, given $\{\alpha_j\}_{j \neq i}$, under the *Simplified Noisy-R* Model satisfies:

$$\exp(-\alpha_i) = \frac{\left( \prod_{(t,c) \in \Phi_i} R_{t,c} \right)^{1/|\Phi_i|}}{\left( \prod_{(t,c) \in \Phi_i} \tilde{R}_{(-i),t,c} \right)^{1/|\Phi_i|}} = \frac{M_0(\{R_{t,c}\}_{\Phi_i})}{M_0(\{\tilde{R}_{(-i),t,c}\}_{\Phi_i})}, \tag{9}$$

where $M_0(\mathcal{S})$ denotes the geometric mean of set $\mathcal{S}$. The ML solution for $\exp(-\alpha_i)$ is the ratio of two geometric means over all country-days when NPI $i$ is active: the numerator is the mean ground-truth $R_{t,c}$ and the denominator is the mean of the predicted value of $R_{t,c}$ if NPI $i$ was deactivated.

To compute the ML solution for the *Simplified Default* Model, recall that Assumption 6 lets us write $\log g(R) = \beta \left( R^{1/\nu} - 1 \right)$, where $\nu$ is the shape and $\beta$ is the inverse scale of the GI distribution, assumed to be a Gamma$(\nu, \beta)$ distribution [2, 7]. We use the well-known analytical form for $M_{\text{GI}}(\cdot)$.

**Theorem 2.** The ML solution of $\alpha_i$, given $\{\alpha_j\}_{j \neq i}$, under the *Simplified Default* Model satisfies:

$$\exp(-\alpha_i) = \left( \sum_{(t,c) \in \Phi_i} \tilde{R}_{(-i),t,c}^{1/\nu} \bar{R}_{t,c}^{1/\nu} \right)^{\nu} \bigg/ \left( \sum_{(t,c) \in \Phi_i} \tilde{R}_{(-i),t,c}^{1/\nu} \tilde{R}_{(-i),t,c}^{1/\nu} \right)^{\nu} = \frac{M_{1/\nu}^{W_i}(\{\bar{R}_{t,c}\}_{\Phi_i})}{M_{1/\nu}^{W_i}(\{\tilde{R}_{(-i),t,c}\}_{\Phi_i})} \tag{10}$$

where $M_{1/\nu}^{W_i}(\mathcal{S})$ is the generalized *weighted* mean of set $\mathcal{S}$, with exponent $1/\nu$ and weights $W_i = \{w_{c,t} = (\tilde{R}_{(-i),t,c})^{\frac{1}{\nu}}\}$. $\bar{R}_{t,c}$ is the ground truth $R$ that exactly corresponds to the observed $g$.

*Proofs:* See Supplement.

Notably, the minor variation in model structure gives a significant difference in ML solutions; the ML solution of the *Simplified Noisy-R* Model is a ratio of geometric means, whilst that of the *Simplified Default* Model is a ratio of generalized weighted means that weighs observations more when the predicted $R$, excluding NPI $i$, is larger. However, in both models, when Assumptions 4 and 2 do not hold, $\alpha_i$ can be interpreted as an average additional effectiveness, since it is produced by averaging over *the data distribution*. Therefore, care must be taken when interpreting NPI effectiveness estimates. For example, we previously estimated that *stay-at-home orders* were associated with a small reduction in $R$ [2]. However, whenever *stay-at-home orders* were active in our data, almost always several other NPIs were also active; consequently, the results should be interpreted as 'implementing a *stay-at-home order* is associated with a modest reduction in $R$ when other effective NPIs are already active'.

## 6 Conclusions

We find that our previously reported NPI effectiveness results [2] are robust across several alternative model structures with transmission noise. For a more comprehensive discussion of the NPI effectiveness results and their implications, we refer the reader to Brauner et al. [2]. While the robustness of these results is promising, the numerous assumptions and limitations inherent to data-driven NPI modelling imply that we should neither treat these results as the last word on NPI effectiveness, nor treat the effects as causal. Instead, policy-makers should draw on diverse sources of evidence, including other retrospective studies, experimental methods, and clinical experience. Our validation suite and model implementations are [available online](#) and we urge those working on estimating NPI effectiveness to systematically validate their models.

## Broader Impact

The rapid pace of the COVID-19 research cycle has increased the erroneous and misreported findings reaching popular attention [21]. It is critical that such errors are caught before publication; the sensitivity analyses developed in this work can uncover faulty assumptions, and so prevent overconfidence or misinformation. We intend for our findings to aid other modelling teams in producing highly reliable, policy-guiding estimates of NPI effects; to this end we release our sensitivity analysis suite and model implementations.

This work is written as many governments are selecting the time and order in which to reintroduce NPIs, and attempting to control second wave epidemics. It offers vital validation of the evidence, to help minimise harm to the world population.

One potential risk stems from miscommunication: we must not mistake high robustness for excessive certainty. We expect the results and conclusions of NPI effectiveness models to change as best practice evolves. In addition, the subtle issues of interpretation raised in Section 5 are difficult to convey to non-technical audiences, and could easily be misread as unconditional effects, or extrapolated incorrectly.

## Acknowledgments

We thank Laurence Aitchison for helpful comments leading to the Additive Effect Model. We thank Tom Rainforth, Eric Nalisnick and Andreas Kirsch for comments on the manuscript.

Mrinank Sharma was supported by the EPSRC Centre for Doctoral Training in Autonomous Intelligent Machines and Systems [EP/S024050/1]. Sören Mindermann's funding for graduate studies was from Oxford University and DeepMind. Jan Brauner was supported by the EPSRC Centre for Doctoral Training in Autonomous Intelligent Machines and Systems [EP/S024050/1] and by Cancer Research UK. Gavin Leech was supported by the UKRI Centre for Doctoral Training in Interactive Artificial Intelligence [EP/S022937/1]. Leonid Chindelevitch acknowledges funding from the MRC Centre for Global Infectious Disease Analysis (reference MR/R015600/1), jointly funded by the UK Medical Research Council (MRC) and the UK Foreign, Commonwealth & Development Office (FCDO), under the MRC/FCDO Concordat agreement and is also part of the EDCTP2 programme supported by the European Union; and acknowledges funding by Community Jameel.

No conflicts of interests.

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
