[Supplementary Material]

# A  Supplementary material for 'How Robust are the Estimated Effects of Nonpharmaceutical Interventions against COVID-19?'

## Contents

## A.1 Additional Model Details

### A.1.1 Model Relationships

**Figure 4:** Relationships between different models.

### A.1.2 Model Assumptions

**Table 1**

| Model | Assumptions |
|-------|-------------|
| Default | Constant epidemiological parameters (1) |
| | No NPI Interactions (4) |
| | Constant NPI Effectiveness (2) |
| | Multiplicative NPI Effects (3) |
| | No Unobserved Factors (5) |
| | Constant exponential growth (6) |
| | Transmission noise (7) |
| | Negative Binomial outputs (8) |
| No Transmission Noise | As Default, except (7) |
| Additive Effects | As Default with (9) instead of (3) |
| Different Effects | As Default with (10) instead of (2) |
| Noisy-R | As Default with Eq.(2.7) instead of (6) |
| Discrete Renewal | As Default with Eq.(2.8) instead of (6) |
| Flaxman et al. [7] | As Discrete Renewal except (7) |

### A.1.3 Default Model Full Description

**Note:** this section is reproduced (with minor modifications) from Brauner et al. [2]. The primary difference is that the model implementations here do not have prior distributions over the parameters of: the generation interval; the delay between infection and case reporting; and the delay between infection and death reporting. These parameters have fixed values set at their prior means — please see Brauner et al. [2] for a detailed justification of parameter values.

Variables are indexed by NPI $i$, country $c$, and day $t$. All prior distributions are independent.

**Data**

1. **NPI Activations**: $x_{i,t,c} \in \{0,1\}$.

2. **Observed (Daily) Cases**: $y_{t,c}^{(C)}$.

3. **Observed (Daily) Deaths**: $y_{t,c}^{(D)}$.

**Prior Distributions**

1. **Country-specific $R_0$:** $R_{0,c} \sim \texttt{Normal}(3.25, \kappa); \quad \kappa \sim \texttt{Half Normal}(\mu = 0, \sigma = 0.5)$.

2. **NPI effectiveness:** $\alpha_i \sim \texttt{Asymmetric Laplace}(m = 0, \kappa = 0.5, \lambda = 10)$. $m$ is the location parameter, $\kappa > 0$ is the asymmetry parameter, and $\lambda > 0$ is the scale parameter.

3. **Infection Initial Counts:**
$$N_{0,c}^{(C)} = \exp(\zeta_c^{(C)}),$$
$$N_{0,c}^{(D)} = \exp(\zeta_c^{(D)}),$$
$$\zeta_c^{(C)} \sim \texttt{Normal}(\mu = 0, \sigma = 50),$$
$$\zeta_c^{(D)} \sim \texttt{Normal}(\mu = 0, \sigma = 50).$$

4. **Observation Noise Dispersion Parameters:**
$$\Psi_{\text{cases}} \sim \texttt{Half Normal}(\mu = 0, \sigma = 5), \tag{11}$$
$$\Psi_{\text{deaths}} \sim \texttt{Half Normal}(\mu = 0, \sigma = 5). \tag{12}$$

**Hyperparameters**

1. **Growth Noise Scale,** $\sigma_g = 0.2$.

**Delay Distributions**

1. **Generation interval distribution** [2]:
$$\mu_{\text{GI}} = 5.06,$$
$$\sigma_{\text{GI}} = 2.11.$$

2. **Time from infection to case confirmation** $\mathcal{T}^{(C)}$ [2]:[2]
$$\mu_{\text{inf}\rightarrow\text{conf}} = 10.92$$
$$\Psi_{\text{inf}\rightarrow\text{conf}} = 5.41.$$
This distribution is converted into a forward-delay vector:
$$\mathcal{T}^{(C)}[t] = \begin{cases} \frac{1}{\mathcal{Z}_C}\texttt{Negative Binomial}(t; \mu = \mu_{\text{inf}\rightarrow\text{conf}}, \alpha = \Psi_{\text{inf}\rightarrow\text{conf}}) & \text{t} < 32 \\ 0 & \text{otherwise} \end{cases},$$
with $\mathcal{Z}_C = \sum_{t'=0}^{31} \texttt{Negative Binomial}(t'; \mu = \mu_{\text{inf}\rightarrow\text{conf}}, \alpha = \Psi_{\text{inf}\rightarrow\text{conf}})$,
i.e., the delay follows a truncated and normalised negative binomial distribution.

3. **Time from infection to death** $\mathcal{T}^{(D)\mathbf{2}}$ [2]:
$$\mu_{\text{inf}\rightarrow\text{death}} = 21.82,$$
$$\Psi_{\text{inf}\rightarrow\text{death}}14.26.$$
This distribution is converted into a forward-delay vector:
$$\mathcal{T}^{(D)}[t] = \begin{cases} \frac{1}{\mathcal{Z}_D}\texttt{Negative Binomial}(t; \mu = \mu_{\text{inf}\rightarrow\text{death}}, \alpha = \Psi_{\text{inf}\rightarrow\text{death}}) & \text{t} < 48 \\ 0 & \text{otherwise} \end{cases},$$
with $\mathcal{Z}_D = \sum_{t'=0}^{47} \texttt{Negative Binomial}(t'; \mu = \mu_{\text{inf}\rightarrow\text{death}}, \alpha = \Psi_{\text{inf}\rightarrow\text{death}})$,
i.e., the delay follows a truncated and normalised negative binomial distribution.

**Infection Model**
$$R_{t,c} = R_{0,c} \cdot \exp\left(-\sum_{i=1}^{I} \alpha_i \, x_{i,t,c}\right), \text{ where } I \text{ is the number of NPIs.}$$
$$\beta_{\text{GI}} = \frac{\mu_{\text{GI}}}{\sigma_{\text{GI}}^2},$$
$$\alpha_{\text{GI}} = \frac{\mu_{\text{GI}}^2}{\sigma_{\text{GI}}^2},$$
$$g_{t,c} = \exp\left(\beta_{\text{GI}}(R_{c,t}^{\frac{1}{\alpha_{\text{GI}}}} - 1)\right) - 1.$$

$$N_{t,c}^{(C)} = N_{0,c}^{(C)} \prod_{\tau=1}^{t} \left[(g_{\tau,c} + 1) \cdot \exp \varepsilon_{\tau,c}^{(C)}\right],$$

$$N_{t,c}^{(D)} = N_{0,c}^{(D)} \prod_{\tau=1}^{t} \left[(g_{\tau,c} + 1) \cdot \exp \varepsilon_{\tau,c}^{(D)}\right], \text{ with noise}$$

$$\varepsilon_{\tau,c}^{(C)} \sim \texttt{Normal}(\mu = 0, \sigma = \sigma_g),$$
$$\varepsilon_{\tau,c}^{(D)} \sim \texttt{Normal}(\mu = 0, \sigma = \sigma_g).$$

**Observation Model**[2]

$$\bar{y}_{t,c}^{(C)} = \sum_{\tau=0}^{31} N_{t-\tau,c}^{(C)} \mathcal{T}^{(C)}[\tau],$$

$$\bar{y}_{t,c}^{(D)} = \sum_{\tau=0}^{47} N_{t-\tau,c}^{(D)} \mathcal{T}^{(D)}[\tau],$$

$$y_{t,c}^{(C)} \sim \texttt{Negative Binomial}(\mu = \bar{y}_{t,c}^{(C)}, \alpha = \Psi^{(C)}),$$

$$y_{t,c}^{(D)} \sim \texttt{Negative Binomial}(\mu = \bar{y}_{t,c}^{(D)}, \alpha = \Psi^{(D)}).$$

### A.2 Holdouts

We evaluate holdout performance by predictive log-likelihood on a test set of 6 countries. We hold out all but the first 14 days of cases and deaths (to allow estimation of $R_{0,c}$ and $N_{0,c}$). Figs 5 and 6 show holdout predictions on this test set. Predictive performance is similar across the models, though models with *transmission noise* tend to perform better. Hyperparameters ($\sigma_g$ or $\sigma_R$, $\sigma_\alpha$) were tuned using 4-fold cross-validation on a previous version of the NPI dataset.

**Figure 5:** Holdout country plots for the *Default, Additive Effects, Different Effects* and *Noisy-R* models.

**Figure 6:** Holdout country plots for the *Discrete Renewal, Deaths-Only Discrete Renewal, Flaxman et al. [8]* and *Default (No Transmission Noise)* models.

## A.3 Full sensitivity results for all models

### A.3.1 Default Model

**Figure 7:** Full sensitivity analysis results for the *Default* model.

### A.3.2  Additive Effects Model

**Figure 8:** Full sensitivity analysis results for the *Additive Effects* model.

## A.3.3 Different Effects Model

**Figure 9:** Full sensitivity analysis results for the *Different Effects* model.

### A.3.4   Noisy-R Model

**Figure 10:** Full sensitivity analysis results for the *Noisy-R* model.

## A.3.5 Discrete Renewal Model

**Figure 11:** Full sensitivity analysis results for the *Discrete Renewal* model.

## A.3.6 Deaths-Only Discrete Renewal Model

**Figure 12:** Full sensitivity analysis results for the *Deaths Only Discrete Renewal* model.

### A.3.7 Flaxman et al. [8] Model

**Figure 13:** Full sensitivity analysis results for the *Flaxman et al. [8]* model.

## A.3.8 Default (No Transmission Noise) Model

**Figure 14:** Full sensitivity analysis results for the *Default (No Transmission Noise)* model.

## A.4    Additional Model Comparison

**Figure 15:** Summarised sensitivity analysis for all models.

## A.5 Discussion of assumptions

We proceed by discussing the assumptions (and their implications) listed in section 2, for which further discussion is necessary.

**Assumption 3** states that each NPI's effect on $R_{t,c}$ is multiplicative. This implies that each NPI has a smaller effect when $R_{t,c}$ is already lowered by other NPIs. Such an assumption may be appropriate because e.g. an active stay-home order decreases the effect of wearing masks in public spaces. However, it may be inappropriate for other NPIs. For example, suppose a given proportion of transmission happens in schools and a given proportion in businesses. In such a situation, closing schools is expected to decrease $R_{t,c}$ by the same amount, whether or not businesses are closed. This leads to an alternative model based on **Assumption 9**, where the effect of each NPI is additive (reprinted from equation (6)):

$$R_{t,c} = R_{0,c} \left( \hat{\alpha} + \sum_{i \in \mathcal{I}} \alpha_i \left( 1 - x_{i,t,c} \right) \right), \quad \text{with } \hat{\alpha} + \sum_{i \in \mathcal{I}} \alpha_i = 1, \tag{13}$$

where the parameter $\hat{\alpha}$ represents the proportion of transmission that still happens when all NPIs are active.

**Assumption 5** states that $R_{t,c}$ depends only on each country's initial reproduction number $R_{0,c}$ and the active NPIs. In other words, no unobserved factors are changing $R_{t,c}$, such as spontaneous social distancing. This is a crucial assumption since the effect of unobserved factors may otherwise be attributed to the active NPIs. This can happen under specific conditions. Firstly, the unobserved effect cannot be present throughout the entire study period since otherwise $R_{0,c}$ accounts for it. Secondly, its timing must be correlated with that of an NPI since otherwise it will be modeled as noise. Under these conditions, an unobserved effect constitutes an *unobserved confounder* [27, 30] or another biasing factor such as a mediator or suppressor. For statistical purposes, there is an equivalence between these types of unobserved effects [25] so we restrict the discussion to confounding.

Without unobserved confounders, our models can infer the *causal* effects of the studied NPIs. This is a property of regression models, such as ours, when their specification is correct [10]. To understand this point intuitively, it is worth examining the simplified models used in section 5.

The effect of unobserved confounders is usually examined by introducing artificial confounders and observing how much this affects results [27, 30]. In the main text, we tested each model's sensitivity to unobserved confounders by making each NPI unobserved, in turn. Results were relatively stable according to the sensitivity loss. However, they are likely to be less stable if there exists a confounder whose effect size and/or correlation with the NPIs exceeds that of the NPIs themselves.

Note that, in principle, it is possible to distinguish changes in $\text{IFR}_c$ and $\text{AR}_c$ from the NPIs' effects: decreasing the ascertainment rate decreases future cases $y_{t,c}^{(C)}$ by a constant factor whereas the introduction of an NPI decreases them by a factor that grows exponentially over time.

## A.6 Proofs of Theorems 1 and 2

*Proof of Theorem 1.* For this model, assume that ground truth values of $R_{t,c}$ have been given to us. By definition, we can write:

$$\log R_{t,c} = \log R_{0,c} - \sum_{i \in \mathcal{I}} \alpha_i \, x_{i,t,c} + \varepsilon_{t,c} \tag{14}$$

where $\varepsilon_{t,c} \sim \mathcal{N}(\mu = 0, \sigma^2 = \sigma_R^2)$; $\sigma_R$ and $R_{0,c}$ are fixed parameters, $x_{i,t,c} \in \{0,1\}$ and $R_{t,c}$ are given. We want to find the maximum likelihood solution for $\{\alpha_i\}_{i \in \mathcal{I}}$.

The log-likelihood $\mathcal{L}$ is given as

$$\mathcal{L} = \sum_{t,c} \log \mathcal{N}(\varepsilon_{t,c}|0, \sigma_R^2) = -\frac{1}{2\sigma_R^2} \sum_{t,c} \varepsilon_{t,c}^2 + \text{constant}, \tag{15}$$

where the constant does not depend on the values of $\{\alpha_i\}_{i \in \mathcal{I}}$. Assume that values $\{\alpha_j\}_{j \in \mathcal{I}, j \neq i}$ are fixed and we are finding the ML solution for $\alpha_i$. Then,

$$\frac{\partial \mathcal{L}}{\partial \alpha_i} \propto \sum_{t,c} \frac{\partial \varepsilon_{t,c}^2}{\partial \alpha_i} \propto \sum_{t,c} \varepsilon_{t,c} x_{i,t,c} = \sum_{(t,c) \in \Phi_i} \varepsilon_{t,c} = \sum_{(t,c) \in \Phi_i} (\log \frac{R_{t,c}}{\tilde{R}_{(-i),t,c}} + \alpha_i), \tag{16}$$

where, as in the main text, $\Phi_i = \{(t,c)|x_{i,t,c} = 1\}$ is the set of days and countries with NPI $i$ active, and $\tilde{R}_{(-i),t,c}$ is the predicted $R$ ignoring the effect of NPI $i$:

$$\tilde{R}_{(-i),t,c} = R_{0,c} \prod_{j \in \mathcal{I} \setminus \{i\}} \exp(-\alpha_j \, x_{j,t,c}) \tag{17}$$

Setting $\frac{\partial \mathcal{L}}{\partial \alpha_i} = 0$, we obtain:

$$-\alpha_i |\Phi_i| = \sum_{(t,c) \in \Phi_i} \log \frac{R_{t,c}}{\tilde{R}_{(-i),t,c}}. \tag{18}$$

By exponentiation and separation into two products, we obtain the theorem statement.

All that remains to show is that $\frac{\partial^2 \mathcal{L}}{\partial \alpha_i^2} < 0$. Preserving signs, but not constants of proportionality, we have:

$$\frac{\partial \mathcal{L}}{\partial \alpha_i} \propto - \sum_{(t,c) \in \Phi_i} \varepsilon_{t,c} \Rightarrow \frac{\partial^2 \mathcal{L}}{\partial \alpha_i^2} \propto - \sum_{(t,c) \in \Phi_i} (1) < 0, \tag{19}$$

as required. $\qquad \square$

*Proof of Theorem 2.* For this model, assume that ground truth values of $g_{t,c}$ have been given to us. Expanding the definitions, we obtain

$$\log g_{t,c} = \beta(R_{0,c}^{1/\nu} \prod_{i \in \mathcal{I}} (\exp(-\alpha_i \, x_{i,t,c})^{1/\nu}) - 1) + \varepsilon_{t,c} \tag{20}$$

where $\varepsilon_{t,c} \sim \mathcal{N}(\mu = 0, \sigma^2 = \sigma_R^2)$; $\sigma_R$, $\nu$, $\beta$ and $R_{0,c}$ are fixed parameters, $x_{i,t,c} \in \{0,1\}$ and $g_{t,c}$ are given.

For each $i \in \mathcal{I}$ independently, we find the maximum likelihood solution $\alpha_i$ given the other $\{\alpha_j\}_{j \in \mathcal{I}, j \neq i}$ in the point where $\partial \mathcal{L}/\partial \alpha_i = 0$. The log-likelihood takes the same form as in Eq. (15). By differentiating, we obtain:

$$\frac{\partial \mathcal{L}}{\partial \alpha_i} \propto - \sum_{t,c} \varepsilon_{t,c} \frac{\partial \varepsilon_{t,c}}{\partial \alpha_i} \tag{21}$$

where we have dropped constants of proportionality but kept the correct signs. Recalling Eq. 20, we can write:

$$\frac{\partial \varepsilon_{t,c}}{\partial \alpha_i} = \frac{\beta}{\nu} \tilde{R}_{t,c}^{1/\nu} x_{i,t,c} \propto \tilde{R}_{t,c}^{1/\nu} x_{i,t,c}. \tag{22}$$

$\tilde{R}_{t,c}$ is the predicted value of $R_{t,c}$ given NPI effectiveness estimates $\{\alpha_i\}_{i\in\mathcal{I}}$ (following Eq. 1 in the main text).

Setting $\frac{\partial \mathcal{L}}{\partial \alpha_i} = 0$ now yields:

$$-\sum_{t,c} \varepsilon_{t,c} x_{i,t,c} \tilde{R}_{t,c}^{1/\nu} = 0 \Rightarrow \exp(-\alpha_i/\nu) \sum_{(t,c)\in\Phi_i} \varepsilon_{t,c} \tilde{R}_{(-i),t,c}^{1/\nu} = 0 \tag{23}$$

Then by expanding $\varepsilon_{t,c}$ using Eq. 20 and expressing $\log g_{t,c}$ in terms of $R_{t,c}$ i.e., converting using Assumption 6, we obtain:

$$\sum_{(t,c)\in\Phi_i} \tilde{R}_{(-i),t,c}^{1/\nu} \left( \beta(\bar{R}_{t,c}^{1/\nu} - 1) - \beta(\tilde{R}_{(-i),t,c}^{1/\nu} \exp(-\alpha_i)^{1/\nu} - 1) \right) = 0. \tag{24}$$

$\bar{R}_{t,c}$ is the value of $R_{t,c}$ produced by converting ground truth values of $g_{t,c}$ using Assumption 6.

From this we obtain the theorem by simplification and rearranging.

All that remains is to show that $\frac{\partial^2 \mathcal{L}}{\partial \alpha_i^2} < 0$. Keeping the signs but dropping constants of proportionality, we have:

$$\frac{\partial \mathcal{L}}{\partial \alpha_i} \propto - \sum_{(t,c)\in\Phi_i} \varepsilon_{t,c} \tilde{R}_{t,c}^{1/\nu}. \tag{25}$$

Therefore:

$$\frac{\partial^2 \mathcal{L}}{\partial \alpha_i^2} \propto - \sum_{(t,c)\in\Phi_i} \left[ \frac{\partial \varepsilon_{t,c}}{\partial \alpha_i} \tilde{R}_{t,c}^{1/\nu} + \varepsilon_{t,c} \frac{\partial \tilde{R}_{t,c}^{1/\nu}}{\partial \alpha_i} \right]$$

$$\propto -\frac{\beta}{\nu} \sum_{(t,c)\in\Phi_i} \left( \tilde{R}_{t,c}^{1/\nu} \right)^2 + \frac{1}{\nu} \underbrace{\sum_{(t,c)\in\Phi_i} \varepsilon_{t,c} \tilde{R}_{t,c}^{1/\nu}}_{\text{0 at ML solution}} \tag{26}$$

Combining Eqs. (21) and (22), we see that the second term is proportional to $\frac{\partial \mathcal{L}}{\partial \alpha_i}$ and therefore 0 at the maximum likelihood solution. Given this, we have $\frac{\partial^2 \mathcal{L}}{\partial \alpha_i^2} < 0$ at $\alpha_i$ satisfying Eq. (24). Therefore, the solution of Eq. (24) is the maximum likelihood solution. $\qquad \square$

## A.7 Experiment Details

### A.7.1 Data Preprocessing

We perform the same data preprocessing as in [2]. To account for the asymmetry between closing and reopening NPIs, our window of analysis terminates 3 days after any NPI is lifted for cases, and 12 days after for deaths. To avoid biasing our models by cases and deaths imported from other countries rather than local cases, we mask cases before a country has reached 100 cumulative cases and deaths before country has reached cumulative 10 deaths. We follow our previous work, and report the *combined* effect of *School Closure* and *University Closure*, since their individual effects cannot be disentangled [2].

### A.7.2 Cross Validation

We previously tuned noise scale hyperparameters on a previous version of our NPI dataset by performing 4 fold cross-validation. We did not update these parameters for the latest dataset. When holding out a country, we do not also hold out the first 14 days of cases and deaths to allow the model to infer $R_{0,c}$, $N_{0,c}^{(C)}$ and $N_{0,c}^{(D)}$. The only way the model is able to explain the remaining held-out data is through these parameters, as well as the shared NPI effectiveness parameters, $\{\alpha_i\}$. We then report predictive likelihood on a test set of 6 countries: Germany, Romania, Mexico, Italy, Austria, Portugal.

### A.7.3 Convergence Statistics

For experiments with default settings, we ensure that $\hat{R} < 1.05$ (i.e., there are no PyMC3 warnings) and that there are no divergent transitions. For the baseline model under default settings, we have $\hat{R} \in [1.000, 1.004]$ for the vast majority of parameters.

### A.7.4 Sensitivity Analyses

We summarise the sensitivity analysis tests we perform here. These are mostly as performed in [2], except that we only perform univariate sensitivity analysis to epidemiological parameters here. Default values are highlighted in bold.

**Sensitivity to Epidemiological Parameters**.

1. We shift the mean infection-to-confirmation delay by $[-3, -1.5, \mathbf{0}, 1.5, +3]$ days.

2. We shift the mean of the infection-to-death distribution by $[-4, -2, \mathbf{0}, +2, +4]$ days.

3. We consider different generation intervals with mean values $[3.06, 4.06, \mathbf{5.06}, 6.06, 7.06]$ days. These distributions have the same standard deviation as the default distribution (2.11 days).

4. We change the mean value of the prior of $R_{0,c}$, $\bar{R}$. We consider values $[2.38, 2.78, \mathbf{3.28}, 3.78, 4.28]$.

5. We change the prior over $\alpha_i$. For all models except the additive model, we try the default asymmetric Laplace prior, $\mathcal{N}(0, 0.2^2)$, $\mathcal{N}^+(0, 0.2^2)$. For the additive model, we use a Dirichlet($\alpha$) prior, where the concentration parameter $\alpha$ is the same for all components. We consider values $[\mathbf{1}, 5, 10]$.

**Data Sensitivity**.

1. We hold out all included countries one at a time.

2. We change the threshold below which confirmed COVID-19 cases are masked in $[10, 30, 50, \mathbf{100}, 200, 300]$.

3. We change the threshold below which COVID-19 deaths are included in $[1, 5, \mathbf{10}, 30, 50]$.

**Sensitivity to Unobserved Factors**.

1. We exclude each of our observed NPIs in turn.

2. We include 5 additional NPIs from the OxCGRT NPI dataset [13]. The NPIs are: *'Travel Screening or Quarantining'* and *'Travel Bans'*; *'Limiting Public Transport Limited'*; *'Limiting Internal Movement Limited'*; *'Public Information Campaigns'* and *'Symptomatic Testing'*.

For sensitivity experiments, we run 4 chains with 1250 samples per chain.

## Footnotes

[2]$\alpha$ in the definition of the Negative Binomial distribution is the dispersion parameter. Larger values of $\alpha$ correspond to a *smaller* variance, and less dispersion. With our parameterisation, the variance of the Negative Binomial distribution is $\mu + \frac{\mu^2}{\alpha}$.