[Reviews · NeurIPS 2020]

Review 1

Summary and Contributions: The paper proposes a few variants of state of the art Bayesian hierarchical models for determining the effectiveness of nonpharmaceutical interventions (NPI). They evaluate the robustness of original models and variants with respect to various assumptions being made in those models.

Strengths: The paper uses a set of extensive evaluations to infer the robustness of various models. The motivation for each assumption for most of the times is well constructed and detailed. The work produces a lot of sensitivity analysis that is of extreme importance and required to understand models in depth.

Weaknesses: The paper essentially compares [2] against [8] along with some variants which have been more or less tested in both [2] and [8]. Hence the contribution of work is essentially checking two assumptions more than previous work. Moreover, [8] has not been implemented or compared to the way proposed by authors. In [8] deaths do not follow a renewal equation only latent infections follow a renewal equation so in Equation[8] just replacing cases with IFR is not correct. The fact that [8] has their code released public it is surprising that authors have implemented their own version which doesn't matches the published version of [8]. Also use using a normal prior with exponential is not correct for estimating effectiveness because it enforces a lot of identifiability issues. These issues arise because of the fact that normal inside an exponential is no more uninformative and importantly it forces a lot of NPIs to have negative effects when many NPIs are colinear or very closely packed. Hence a better choice is to use priors which give an equal chance to each NPI to have a positive effect or no effect and combined them together they should have a uniform chance of decreasing Rt by 0 to 100%. Hence there is a need to carefully extend models by giving a thought about what parameters stand for rather doing an abaltion study. Also [8] has an option to have all the NPIs having a region/country specific effect so top test that assumption a model tweak to do that should be compared. Moreover, studies like removing a NPI is not ideal in way that you have already had impact of that NPI so removing it from data doesn't make sense as the effect on NPI has already been observed or reflected in data so now running it without this would mean attributing it to other NPI.

Correctness: The claims per se are correct but the fact that one of the methods has not been implemented the inteded way makes most of the claims invalidated.

Clarity: Paper is well written and motivation behind assumptions are defined properly.

Relation to Prior Work: Prior work has been discussed well, however it would be good if authors can at least comment on mechanistic models used for NPI effectiveness for sake of completeness.

Reproducibility: Yes

Additional Feedback: After reading the rebuttal I am increasing my score to 7. However, there are still a few points that need more clarification in the manuscript (hope it happens). 1. Thanks for pointing out the Soletz et al paper. But it seems they do not talk about effectiveness of NPIs using the same model as Flaxman et.al. The model criticized by them is essentially a modification of Flaxman et.al model which doesn't do partial pooling, admittedly even the model presented in this paper will fail for that. But that doesn't invalidate this model, the model by Soletz et al is purely restrictive. So either present same analysis on your model where you remove partial effect sizes or just talk about Soletz et al using a restritive model. 2. Even the comparison done here in the paper are essentially done using more data than Falxman et.al, which now allows you to use partial pooling of all covariates. So better use a modification of their model by using partial pooling for all covariates and then present the analysis. Else it feels a bit unjustified comparison 3. Still I don't get how can Theorem is any useful? No way you will ever get the ground truth R0/Rt of the disease. 4. Also, Flaxman et.al in their supplementary do not present the sensitivity on various options is a wrong claim. They do and present only things that changed and note that nothing else changes, so not producing a figure is not indicative of not performing an analysis, just that they reported what changed and rest was claimed as nothing changed.


Review 2

Summary and Contributions: This paper performs a sensitivity analysis of the modeling choices used to estimate the effects of nonpharmaceutical interventions to reduce COVID-19 spread.

Strengths: Evidence for or against the effectiveness of nonpharmaceutical interventions to reduce COVID-19 spread are a high priority in research today. Sensitivity analysis in this setting is critical to understand whether data, assumptions, or both are providing this evidence. The paper aims to decompose these in an important way.

Weaknesses: The literature on policy or program evaluation and epidemiology emphasizes the importance of the no unobserved confounder assumption in estimating effectiveness of interventions from data. In the problem studied in this paper, confounders would be differences between regions that make them more likely to use a given intervention and also have different epidemiological parameters regarding the spread of COVID-19. However, this paper does not perform any sensitivity analysis with respect to this assumption. The assumptions tested, a priori, are unlikely to effect the conclusions, and therefore the paper does not provide strong evidence for their top-line claim that "all high-level conclusions are highly robust." Edit: Thanks to the authors for adding a sensitivity analysis for confounding. I am familiar with Rosenbaum's sensitivity analysis for unobserved confounders, and think it is a reasonable and appropriate choice here.

Correctness: There is significant over-reach in the conclusion that these models are robust to epidemiological assumptions. Edit: Thanks to the authors for agreeing to tone down the over-reach.

Clarity: The paper is somewhat disorganized, the story is not fully put together and reads as a collection of results rather than a coherent insight into better understanding of the current evidence regarding nonpharmaceutical interventions.

Relation to Prior Work: Yes, it is well contextualized in the COVID-19 and infectious disease literature, although it lacks some contextualization in the causal inference or program evaluation literature on which much of this work depends. Edit: including a discussion of Rosenbaum's sensitivity analysis is a good step in the right direction. Clear contextualization in the causal literature is important for this work.

Reproducibility: Yes

Additional Feedback:


Review 3

Summary and Contributions: The authors perform a careful empirical study, to assess how modeling assumptions impact inference of the role of non-pharmaceutical intervention (NPI) in mitigating covid-19 transmissions. Their contributions can be summed as: 1) writing out and summarizing in one place the current set of epidemiological models used for forecasting covid-19 transmission and growth as a function of NPIs and other parameters, 2) careful analysis of impact of model assumptions, model fit and parameter robustness on results.

Strengths: Overall, I wasn’t convinced that NeurIPS is the right venue for this paper, given its lack of novelty on methods, data, or model/parameter assessment. However, given the importance of this problem, I think it still could provide value by introducing in one place a range of models to NeurIPS audience. But, several major criticism describe below significantly dampen my enthusiasm for this submission.

Weaknesses: + The authors are re-driving, and relating, the standard epidemiological models used for modeling transmission rate and other related parameters. However, as the author also point out, all these models are making assumptions that we now know are violated. For instance, “school closure” is too broad of the category, given the known differences in transmission rates across age groups (e.g., Chikina M, Pegden W, 2020). As another example, death rate varies by age group, and countries have very different age distributions. Instead of using “standard” approaches that makes such simplifying assumptions (and trying to fit the data to the same old models), I much rather see new models that can handles what we know about this particular pandemic. If there are not enough data to move in that direction, then a fitting question would be how much data is needed for instance through simulation and power analysis. + Measuring test set likelihood is good for model comparison, but overall hard to understand how “good” a model is. It would be more informative to see results on some sort of “null data” for instance, by fitting and assessing models on data where interventions across countries are permuted. + If we consider the manuscript for the results, as opposed to innovative models (that work better than baseline), then the next bar would be novelty of results. The data is taken exactly from [2], and no new insight is gained (beyond the observation of robustness of results across a range of assumptions -- [2] already did a good job of assessing parameter sensitivity). + Confounding factors (e.g., to what extend a population actually respects an intervention) is a major problem for reasoning about effectiveness of NPIs. The authors have attempted to asses this by assessing country-specific effectiveness (Figure 3) of various NPI. However, it’s hard to interpret the credibility of these results. The authors should go one step further to attempt to “explain” these estimate by other country-specific factors, in order to at least anecdotally provide more evidence for these observations. E.g., can you correlate the differences in “mask wearing” effectiveness across countries to data such as similarity of demographics etc.

Correctness: Yes

Clarity: Paper is well written, however, I found that many of the critical pieces to fully understand the methods were in the supplement. Given that this is submitted to an ML conference, it would be more fitting to fully describe the model in the main methods (including what are known parameters and which ones are estimated by algorithm)

Relation to Prior Work: yes

Reproducibility: Yes

Additional Feedback: See my comments on "Weaknesses"

[Author Response · NeurIPS 2020]

We thank the reviewers for their constructive feedback. We appreciate the comments that our "careful empirical study" **[R4]** and "sensitivity analyses [...] are of extreme importance" **[R2]**. Further, they are "critical to understand[ing] whether assumptions, data or both are providing evidence" **[R3]** about the effectiveness of different nonpharmaceutical interventions (NPIs) against COVID19 transmission. Given the importance and time-sensitivity of these results, and the minor criticisms raised by the reviewers, we hope that our clarifications below will allow the reviewers to increase their scores. Additionally, in line with reviewer comments, **we have run a number of additional experiments that will also be included in the camera-ready version**.

**[R2] [R3] [R4] Contribution.** Our work is the first that performs *structural* sensitivity analysis and compares the robustness of data-driven NPI effectiveness models. Our findings are policy relevant; the high sensitivity of the model used in [8], subsequently published in Nature, raises concerns (though the authors do not claim to distinguish individual NPI effects). A recent preprint (concurrent work to us) also finds [8] has high sensitivity [*Soltesz et al, On the sensitivity of non-pharmaceutical intervention models for SARS-CoV-2 spread estimation, 2020*]. We highlight that neither [2] nor [8] test structural assumptions[1], and [8] *never reports the sensitivity of NPI effectiveness estimates in the tests they perform.* **[R2]** correctly points out that these models make "assumptions that we now know are violated", exactly why our novel mathematical results (§5 *Effectiveness in Context*) are important steps forward. Our results show that when commonly made assumptions are violated, estimates must be interpreted as averages, taken over contexts of the dataset, and expert judgement is required to adjust them to local, unique circumstances.

**[R2] Implementation.** Our implementation of the model of [8] is correct; we only model latent infections as a discrete renewal process while deaths are modelled as in [8] and [2] i.e., produced via discrete convolutions. **We believe this misunderstanding is due to typographical errors in the supplement, Eqs. (128), (129)**. The correct equation, modelling only deaths, is $N_{t,c}^{(D)} = R_{t,c} \sum_{\tau=1}^{t} N_{t-\tau,c}^{(D)} \cdot \pi_{SI}[\tau]$ where $\pi_{SI}[\tau]$ is the discretised serial interval distribution, $N_{t,c}^{(D)}$ is the daily number of infections that result in fatalities. $R_{t,c}$ is the instantaneous reproduction number at time $t$ in country $c$. We seed this with a latent variable $N_{0,c}^{(D)}$ that incorporates the country-specific

Figure 1: *Additional experiments using the baseline model. Top: **[R2]** prior from [8]. Bottom: **[R3] [R4]** additional confounding tests. The NPIs from the OxCGRT dataset (as labeled) are now observed.*

infection fatality rate, $\text{IFR}_c$. Other than truncation and naming, this is identical to $c_{t,m} = R_{t,m} \sum_{\tau=0}^{t-1} c_{\tau,m} g_{t-t}$ [8] where the convolution has been rewritten indexing over the other variable. Since $c$ represents the *total* number of infections, we have $c_{t,c} = N_{t,c}^{(D)}/\text{IFR}_c$. We compute the expected number of deaths as $\bar{D}_{t,c} = \sum_{\tau=1}^{63} N_{t-\tau,c}^{(D)} \pi_D[\tau]$, where $\pi_D[\tau]$ is the discretised infection-to-death delay. We implemented all models ourselves to minimise discrepancies between models and make fair comparisons.

**[R3] [R4] Confounding.** Thank you for pointing out the no-confounder assumption. We agree that this assumption is critical, and will update the tone of the conclusion to reflect the assumptions we tested. For clarity, the NPI leave-out test assesses how much the effect of unobserved interventions are attributed to observed NPIs **[R2] thereby testing this assumption**. We apologise for not clarifying the purpose of this test. In light of your feedback, we have run additional experiments finding low sensitivity when previously unobserved NPIs from the OxCGRT NPI dataset [*Thomas Hale et al. Oxford COVID-19 Government Response Tracker. (2020)*] are observed (Fig. 1, bottom). These tests are imperfect but considered best practice [*Rosenbaum et al., Assessing Sensitivity to an Unobserved Binary Covariate in an Observational Study with Binary Outcome, 1983*]. We highlight that our results show that confounding is the key limitation of such NPI effectiveness models. For example, if we had found that effectiveness estimates fluctuate widely under different epidemiological parameters, we would not have been able to make strong conclusions regardless of whether we observe all relevant factors.

**[R2] Effectiveness Prior.** Thank you for your comment. We take our effectiveness prior from [2], and it reflects the belief that NPIs have moderate effects. Low posterior correlation $r < 0.4$ between NPI effectiveness estimates and low sensitivity suggests that collinearity is manageable. Furthermore, we have added run a test using the suggested prior from [8] (Fig. 1, top).

## Footnotes

[1]The most recent version of [2] reproduces our structural sensitivity analysis from the preprint corresponding to this submission.


[Meta-Review · NeurIPS 2020]

This paper presents novel analyses using hierarchical Bayesian models to estimate the effectiveness of nonpharmaceutical interventions in order to reduce COVID-19 spread. All reviewers agreed on the significance of the analyses, and major concerns and questions were resolved by author feedback very successfully. Given the extraordinary importance of research on this subject at present times, we have explicitly called out in our call for papers. Therefore, this paper is above the bar for acceptance to NeurIPS.